# Convergence Analysis of Schrödinger-Föllmer Sampler without Convexity

## Abstract

Schrödinger-Föllmer sampler (SFS) (Huang et al., 2021) is a novel and efficient approach for sampling from possibly unnormalized distributions without ergodicity. SFS is based on the Euler-Maruyama discretization of Schrödinger-Föllmer diffusion process

$$\mathrm{d}X_t = -\nabla U\left(X_t, t\right)\mathrm{d}t + \mathrm{d}B_t, \quad t \in [0, 1], \quad X_0 = 0$$

on the unit interval, which transports the degenerate distribution at time zero to the target distribution at time one. In Huang et al. (2021), the consistency of SFS is established under a restricted assumption that the potential $U(x, t)$ is uniformly (on $t$) strongly convex (on $x$). In this paper we provide a non-asymptotic error bound of SFS in Wasserstein-2 distance under some smooth and bounded conditions on the density ratio of the target distribution over the standard normal distribution, but without requiring strong convexity of the potential.

## 1 Introduction

Sampling from possibly unnormalized distributions is an important task in Bayesian statistics and machine learning. Ever since the Metropolis-Hastings (MH) algorithm (Metropolis et al., 1953; Hastings, 1970) was introduced, various random sampling methods were proposed, including Gibbs sampler, random walk sampler, independent sampler, Langevin sampler, bouncy particle sampler, zig-zag sampler (Geman & Geman, 1984; Gelfand & Smith, 1990; Tierney, 1994; Liu, 2008; Robert et al., 2010; Bouchard-Côté et al., 2018; Bierkens et al., 2019), among others, see Brooks et al. (2011); Dunson & Johndrow (2020) and the references therein. The above mentioned sampling algorithms generate random samples by running an ergodic Markov chain whose stationary distribution is the target distribution.

In Huang et al. (2021), the Schrödinger-Föllmer sampler (SFS), a novel sampling approach without requiring the property of ergodicity is proposed. SFS is based on the Schrödinger-Föllmer diffusion process, defined as

$$\mathrm{d}X_t = b\left(X_t, t\right)\mathrm{d}t + \mathrm{d}B_t, \quad t \in [0, 1], \quad X_0 = 0, \tag{1}$$

where the drift function

$$b(x, t) = -\nabla U(x, t) = \frac{\mathbb{E}_{Z \sim N(0, \boldsymbol{I}_p)}[\nabla f(x + \sqrt{1-t}Z)]}{\mathbb{E}_{Z \sim N(0, \boldsymbol{I}_p)}[f(x + \sqrt{1-t}Z)]} : \mathbb{R}^p \times [0, 1] \to \mathbb{R}^1$$

with $f(\cdot) = \frac{d\mu}{dN(0, \boldsymbol{I}_p)}(\cdot)$. Here, we assume that $f(\cdot)$ is twice differentiable. According to Léonard (2014) and Eldan et al. (2020), the process $\{X_t\}_{t \in [0, 1]}$ in (1) was first formulated by Föllmer (Föllmer, 1985; 1986; 1988) when studying the Schrödinger bridge problem (Schrödinger, 1932). The main feature of the above Schrödinger-Föllmer process is that it interpolates $\delta_0$ and the target $\mu$ in time $[0, 1]$, i.e., $X_1 \sim \mu$, see Proposition 2.1. SFS samples from $\mu$ via the following Euler-Maruyama discretization of (1),

$$Y_{t_{k+1}} = Y_{t_k} + sb\left(Y_{t_k}, t_k\right) + \sqrt{s}\epsilon_{k+1}, \ Y_{t_0} = 0, \ k = 0, 1, \ldots, K-1, \tag{2}$$

where $s = 1/K$ is the step size, $t_k = sk$, and $\{\epsilon_k\}_{k=1}^{K}$ are independent and identically distributed from $N(0, \mathbf{I}_p)$. If the expectations in the drift term $b(x, t)$ do not have analytical forms, one can use Monte Carlo method to evaluate $b(Y_{t_k}, t_k)$ approximately, i.e., one can sample from $\mu$ according

$$\widetilde{Y}_{t_{k+1}} = \widetilde{Y}_{t_k} + s\tilde{b}_m\left(\widetilde{Y}_{t_k}, t_k\right) + \sqrt{s}\epsilon_{k+1}, \ \widetilde{Y}_{t_0} = 0, \ k = 0, 1, \ldots, K-1,$$

where $\tilde{b}_m(\widetilde{Y}_{t_k}, t_k) = \frac{\frac{1}{m}\sum_{j=1}^{m}[\nabla f(\widetilde{Y}_{t_k} + \sqrt{1-t_k}Z_j)]}{\frac{1}{m}\sum_{j=1}^{m}[f(\widetilde{Y}_{t_k} + \sqrt{1-t_k}Z_j)]}$ with $Z_1, \ldots, Z_m$ i.i.d $N(0, \mathbf{I}_p)$. The numerical simulations in Huang et al. (2021) demonstrate that SFS outperforms the existing samplers based on ergodicity including Langevin-based samplers, see (Huang et al., 2021, Section 3) for a detailed discussion on the comparison with Langevin-based samplers.

In Section 4.2 of Huang et al. (2021), they prove that

$$W_2(\text{Law}(\widetilde{Y}_{t_K}), \mu) \to 0, \quad \text{as} \quad s \to 0, m \to \infty$$

under a restricted assumption that the potential $U(x, t)$ is uniformly strongly convex in $x$, i.e.,

$$U(x, t) - U(y, t) - \nabla U(y, t)^\top (x - y) \geq (M/2) \|x - y\|_2^2, \forall x, y \in \mathbb{R}^p, \forall t \in [0, 1], \tag{3}$$

where $M$ is a positive constant. In this paper we provide a new analysis of the above SFS iteration. We establish a non-asymptotic error bound on $W_2(\text{Law}(\widetilde{Y}_{t_K}), \mu)$ under the condition that $f$ and $\nabla f$ are Lipschitz continuous and $f$ has positive lower bound, but without using the uniform strong convexity requirement (3).

The rest of this paper is organized as follows. In Section 2, we recall the SFS method. In Section 3, we present our theoretical analysis. We conclude in Section 4. Proofs for all the theorems are provided in Appendix.

## 2 Schrödinger-Föllmer sampler

In this section we recall the Schrödinger-Föllmer sampler briefly. More background on the Schrödinger-Föllmer diffusion process please see Dai Pra (1991); Léonard (2014); Chen et al. (2021); Huang et al. (2021).

Let $\mu \in \mathcal{P}(\mathbb{R}^p)$ be the target distribution and absolutely continuous with respect to the $p$-dimensional standard normal measure $G = N(0, \mathbf{I}_p)$, where $\mathcal{P}(\mathbb{R}^d)$ refers to the probability measures defined on the Borel space $(\mathbb{R}^d, \mathcal{B}(\mathbb{R}^d))$. Let

$$f(x) = \frac{d\mu}{dG}(x).$$

We assume that

    (**A1**) $f, \nabla f$ are Lipschitz continuous with constant $\gamma$,

    (**A2**) There exists $\xi > 0$ such that $f \geq \xi$.

Define the heat semigroup $Q_t, t \in [0, 1]$ as

$$Q_t f(x) = \mathbb{E}_{Z \sim G}[f(x + \sqrt{t}Z)].$$

**Proposition 2.1.** *Define a drift function*

$$b(x, t) = \nabla \log Q_{1-t}f(x).$$

*If $f$ satisfies assumptions (**A1**) and (**A2**), then the Schrödinger-Föllmer diffusion*

$$\mathrm{d}X_t = b(X_t, t)\,\mathrm{d}t + \mathrm{d}B_t, \quad t \in [0, 1], \quad X_0 = 0, \tag{4}$$

*has a unique strong solution and $X_1 \sim \mu$.*

**Remark 2.1.**

(i) (**A1**) and (**A2**) directly follow from Assumption 1 of Tzen & Raginsky (2019), Definition 5 and Lemma 6 of Lehec (2013).

(ii) From the definition of $b(x,t)$ in Proposition 2.1, it follows that $U(x,t) = -\log Q_{1-t}f(x)$.

(iii) If the target distribution is $\mu(dx) = \exp(-V(x))dx/C$ with the normalized constant $C$, then $f(x) = \frac{(\sqrt{2\pi})^p}{C}\exp\left(-V(x) + \frac{\|x\|_2^2}{2}\right)$. If $V(x)$ is twice differentiable and

$$\lim_{R\to\infty}\sup_{\|x\|_2\geq R}\exp\left(-V(x) + \|x\|_2^2/2\right)\|x - \nabla V(x)\|_2 < \infty,$$

$$\lim_{R\to\infty}\sup_{\|x\|_2\geq R}\exp\left(-V(x) + \|x\|_2^2/2\right)\|\boldsymbol{I}_p - \nabla^2 V(x)\|_2 < \infty,$$

then both $f$ and $\nabla f$ are Lipschitz continuous, i.e., (**A1**) holds. (**A2**) is equivalent to the growth condition on the potential that $V(x) \leq \frac{\|x\|^2}{2} - \log\xi + \text{constant}$. In this case, any potential $V$ taking in the quadratic form can be considered to satisfy conditions (**A1**) and (**A2**), since its corresponding random variable can be transformed into the standard Gaussian random variable by scaling.

(iv) Under (**A1**) and (**A2**), some calculation shows that

$$\sup_{x\in\mathbb{R}^p}\|\nabla f(x)\|_2 \leq \gamma, \quad \sup_{x\in\mathbb{R}^p}\|\nabla^2 f(x)\|_2 \leq \gamma,$$

and

$$\sup_{x\in\mathbb{R}^p, t\in[0,1]}\|\nabla Q_{1-t}f(x)\|_2 \leq \gamma, \quad \sup_{x\in\mathbb{R}^p, t\in[0,1]}\|\nabla^2(Q_{1-t}f(x))\|_2 \leq \gamma,$$

and

$$b(x,t) = \frac{\nabla Q_{1-t}f(x)}{Q_{1-t}f(x)}, \quad \nabla b(x,t) = \frac{\nabla^2(Q_{1-t}f(x))}{Q_{1-t}f(x)} - b(x,t)b(x,t)^\top.$$

We conclude that

$$\sup_{x\in\mathbb{R}^p, t\in[0,1]}\|b(x,t)\|_2 \leq \frac{\gamma}{\xi}, \quad \sup_{x\in\mathbb{R}^p, t\in[0,1]}\|\nabla b(x,t)\|_2 \leq \frac{\gamma}{\xi} + \frac{\gamma^2}{\xi^2}.$$

Proposition 2.1 shows that the Schrödinger-Föllmer diffusion will transport $\delta_0$ to the target $\mu$ on the unite time interval. Since the drift term $b(x,t)$ is scale-invariant with respect to $f$ in the sense that $b(x,t) = \nabla\log Q_{1-t}Cf(x), \forall C > 0$. Therefore, the Schrödinger-Föllmer diffusion can be used for sampling from $\mu(dx) = \exp(-V(x))dx/C$, where the normalizing constant of $C$ may not be known. In that case, we use the Euler-Maruyama method to discretize the Schrödinger-Föllmer diffusion (4). Let

$$t_k = k\cdot s, \quad k = 0, 1, \ldots, K, \quad \text{with} \quad s = 1/K, \quad Y_{t_0} = 0,$$

the Euler-Maruyama scheme reads (2), among which the drift term is explicitly expressed as

$$b(Y_{t_k}, t_k) = \frac{\mathbb{E}_Z[\nabla f(Y_{t_k} + \sqrt{1-t_k}Z)]}{\mathbb{E}_Z[f(Y_{t_k} + \sqrt{1-t_k}Z)]} = \frac{\mathbb{E}_Z[Zf(Y_{t_k} + \sqrt{1-t_k}Z)]}{\mathbb{E}_Z[f(Y_{t_k} + \sqrt{1-t_k}Z)]\sqrt{1-t_k}}. \tag{5}$$

In (5), the second equality follows from Stein's lemma (Stein et al., 1972; Stein, 1986; Landsman & Nešlehová, 2008). From the definition of $b(Y_{t_k}, t_k)$ in (5), we may not get its explicit expression. Here, we can get one estimator $\tilde{b}_m$ of $b$ by replacing $\mathbb{E}_Z$ in $b$ with $m$-sample mean, i.e.,

$$\tilde{b}_m(Y_{t_k}, t_k) = \frac{\frac{1}{m}\sum_{j=1}^m[\nabla f(Y_{t_k} + \sqrt{1-t_k}Z_j)]}{\frac{1}{m}\sum_{j=1}^m[f(Y_{t_k} + \sqrt{1-t_k}Z_j)]}, \quad k = 0, \ldots, K-1, \tag{6}$$

or

$$\tilde{b}_m(Y_{t_k}, t_k) = \frac{\frac{1}{m}\sum_{j=1}^m [Z_j f(Y_{t_k} + \sqrt{1-t_k}Z_j)]}{\frac{1}{m}\sum_{j=1}^m [f(Y_{t_k} + \sqrt{1-t_k}Z_j)] \cdot \sqrt{1-t_k}}, \quad k = 0, \ldots, K-1, \tag{7}$$

where $Z_1, \ldots, Z_m$ are i.i.d. $N(0, \boldsymbol{I}_p)$. The detailed description of SFS is summarized in following Algorithm 1 below, which is Algorithm 2 in Huang et al. (2021).

---

**Algorithm 1** SFS for $\mu = \exp(-V(x))/C$ with Monte Carlo estimation of the drift term

---

Input: $m$, $K$. Initialize $s = 1/K$, $\widetilde{Y}_{t_0} = 0$.
**for** $k = 0, 1, \ldots, K-1$ **do**

    Sample $\epsilon_{k+1} \sim N(0, \boldsymbol{I}_p)$.

    Sample $Z_i, i, \ldots, m$, from $N(0, \boldsymbol{I}_p)$.

    Compute $\tilde{b}_m$ according to (6) or (7).

    $\widetilde{Y}_{t_{k+1}} = \widetilde{Y}_{t_k} + s\tilde{b}_m\left(\widetilde{Y}_{t_k}, t_k\right) + \sqrt{s}\epsilon_{k+1}$.

**end for**
Output: $\{\widetilde{Y}_{t_k}\}_{k=1}^K$.

---

In Section 4.2 of Huang et al. (2021), they proved that

$$W_2(\text{Law}(\widetilde{Y}_{t_K}), \mu) \to 0, \quad \text{as} \quad s \to 0, m \to \infty$$

under the uniform strong convexity assumption (3). However, (3) is not easy to verify. In the next section, we establish a nonasymptotic bound on the Wasserstein-2 distance between the law of $\widetilde{Y}_{t_K}$ generated by SFS (Algorithm 1) and the target $\mu$ under smooth and bounded conditions (**A1**) and (**A2**) but without using (3).

## 3  Bound on $W_2(\text{Law}(\widetilde{Y}_{t_K}), \mu)$ without convexity

Under conditions (**A1**) and (**A2**), one can easily deduce the growth condition and Lipschitz/Hölder continuity of the drift term $b(x, t)$ (Huang et al., 2021, Remark 4.1), i.e.,

$$\|b(x, t)\|_2^2 \leq C_0(1 + \|x\|_2^2), \tag{C1}$$

and

$$\|b(x, t) - b(y, t)\|_2 \leq C_1 \|x - y\|_2, \tag{C2}$$

and

$$\|b(x, t) - b(y, s)\|_2 \leq C_1\left(\|x - y\|_2 + |t - s|^{\frac{1}{2}}\right), \tag{C3}$$

where $C_0$ and $C_1$ are two positive constants. See Section A.3 in Huang et al. (2021) for the detailed derivations of (C1)-(C3).

**Remark 3.1.** *(C1) and (C2) are the essentially sufficient conditions such that the Schrödinger-Föllmer SDE (4) admits a unique strong solution. (C3) has been introduced in Theorem 4.1 of Tzen & Raginsky (2019), and it is also similar to the condition H2 of Chau et al. (2021) and Assumption 3.2 of Barkhagen et al. (2021). Obviously, (C3) implies (C1) and (C2) hold if the drift term $b(x, t)$ is bounded over $\mathbb{R}^p \times [0, 1]$.*

Let $\mathcal{D}(\nu_1, \nu_2)$ be the collection of coupling probability measures on $\left(\mathbb{R}^{2p}, \mathcal{B}(\mathbb{R}^{2p})\right)$ such that its respective marginal distributions are $\nu_1$ and $\nu_2$. The Wasserstein-2 distance with which we measure the discrepancy between $\text{Law}(\widetilde{Y}_{t_K})$ and $\mu$ is defined as

$$W_2(\nu_1, \nu_2) = \inf_{\nu \in \mathcal{D}(\nu_1, \nu_2)} \left( \int_{\mathbb{R}^p} \int_{\mathbb{R}^p} \|\theta_1 - \theta_2\|_2^2 \, d\nu(\theta_1, \theta_2) \right)^{1/2}.$$

**Theorem 3.1.** *Assume* **(A1)** *and* **(A2)** *hold, then*

$$W_2(Law(\widetilde{Y}_{t_K}), \mu) \leq \mathcal{O}(\sqrt{ps}) + \mathcal{O}\left( \sqrt{\frac{p}{\log(m)}} \right),$$

*where $s = 1/K$ is the step size.*

**Remark 3.2.** *This theorem provides some guidance on the selection of $s$ and $m$. To ensure convergence of the distribution of $\widetilde{Y}_{t_K}$, we should set the step size $s = o(1/p)$ and $m = \exp(p/o(1))$. In high-dimensional models with a large $p$, we need to generate a large number of random vectors from $N(0, \boldsymbol{I}_p)$ to obtain an accurate estimate of the drift term $b$. If we assume that $f$ is bounded above we can improve the nonasymptotic error bound, in which $\mathcal{O}\left(\sqrt{p/\log(m)}\right)$ can be improved to be $\mathcal{O}\left(\sqrt{p/m}\right)$.*

**Theorem 3.2.** *Assume that, in addition to the conditions of Theorem 3.1, $f$ has a finite upper bound, then*

$$W_2(Law(\widetilde{Y}_{t_K}), \mu) \leq \mathcal{O}(\sqrt{ps}) + \mathcal{O}\left( \sqrt{\frac{p}{m}} \right),$$

*where $s = 1/K$ is the step size.*

**Remark 3.3.** *With the boundedness condition on $f$, to ensure convergence of the sampling distribution, we can set the step size $s = o(1/p)$ and $m = p/o(1)$. Note that the sample size requirement for approximating the drift term is significantly less stringent than that in Theorem 3.1.*

**Remark 3.4.** *Langevin sampling method has been studied under the (strongly) convex potential assumption (Durmus & Moulines, 2019; 2016; 2017; Dalalyan, 2017a;b; Cheng & Bartlett, 2018; Dalalyan & Karagulyan, 2019); the dissipativity condition for the drift term (Raginsky et al., 2017; Mou et al., 2022; Zhang et al., 2019); the local convexity condition for the potential function outside a ball (Durmus & Moulines, 2017; Cheng et al., 2018; Ma et al., 2019; Bou-Rabee et al., 2020). Moreover, the constant in the log Sobolev inequality depends on the dimensionality exponentially (Wang et al., 2009; Hale, 2010; Menz et al., 2014; Raginsky et al., 2017), implying that the Langevin samplers suffers from the curse of dimensionality. SFS does not require the underlying Markov process to be ergodic, therefore, our results in Theorems 3.1-3.2 are established under the smooth and bounded assumptions (A1) and (A2) on $f$ but do not need the above mentioned conditions used in the analysis of Langevin samplers.*

In Theorems 3.1-3.2, we use (**A2**), i.e, $f$ has positive lower bound, however, (**A2**) may not hold if the target distribution admits compact support. To circumvent this difficulty, we consider the regularized probability measure

$$\mu_\varepsilon = (1 - \varepsilon)\mu + \varepsilon G, \quad \varepsilon \in (0, 1).$$

The corresponding density ratio is

$$f_\varepsilon = \frac{d\mu_\varepsilon}{dG} = (1 - \varepsilon)f + \varepsilon.$$

Obviously, $f_\varepsilon$ satisfies (**A1**) and (**A2**) if $f$ and $\nabla f$ are Lipschitz continuous. Since $\mu_\varepsilon$ can approximate to $\mu$ well if we set $\varepsilon$ small enough, then we consider sampling from $\mu_\varepsilon$ by running SFS (Algorithm 1) with $f$ being replaced by $f_\varepsilon$. We use $\widetilde{Y}_{t_K}^\varepsilon$ to denote the last iteration of SFS.

**Theorem 3.3.** *Assume* **(A1)** *holds and $\varepsilon = (\log(m))^{-1/5}$, then*

$$W_2(Law(\widetilde{Y}_{t_K}^\varepsilon), \mu) \leq \mathcal{O}(\sqrt{ps}) + \widetilde{C}_p \cdot \mathcal{O}\left( \frac{1}{(\log(m))^{1/10}} \right),$$

where $s = 1/K$ is the step size, $\widetilde{C}_p$ is a constant depending on $p$. Moreover, if $f$ has a finite upper bound and $\varepsilon = (\log(m))^{-1/5}$, then

$$W_2(Law(\widetilde{Y}_{t_K}^\varepsilon), \mu) \leq \mathcal{O}(\sqrt{ps}) + \widetilde{C}_p \cdot \mathcal{O}\left(\frac{1}{m^{1/10}}\right).$$

## 4    Conclusion

In Huang et al. (2021), Schrödinger-Föllmer sampler (SFS) was proposed for sampling from possibly unnormalized distributions. The key feature of SFS is that it does not need ergodicity as its theoretical basis. The consistency of SFS proved in Huang et al. (2021) relies on a restricted assumption that the potential function is uniformly strongly convex. In this paper we provide a new convergence analysis of the SFS without the strongly convexity condition on the potential. We establish a non-asymptotic error bound on Wasserstein-2 distance  between the law of the output of SFS and the target distribution under smooth and bounded assumptions on the density ratio of the target distribution over the standard normal distribution.

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

## A  Appendix

In this appendix, we prove Proposition 2.1 and Theorems 3.1-3.3.

### A.1  Proof of Proposition 2.1

*Proof.* This is a known result, see Dai Pra (1991); Lehec (2013) for details. □

### A.2  Preliminary lemmas for Theorems 3.1-3.2

First, recall that the Schrödinger-Föllmer diffusion in (4) is defined as

$$\mathrm{d}X_t = b\left(X_t, t\right)\mathrm{d}t + \mathrm{d}B_t, \quad t \in [0, 1], \quad X_0 = 0, \ X_1 \sim \mu.$$

Then we introduce Lemmas A.1-A.5 in preparing for the proofs of Theorems 3.1-3.2.

**Lemma A.1.** *Assume (**A1**) and (**A2**) hold, then*

$$\mathbb{E}[\|X_t\|_2^2] \le 2(C_0 + p)\exp(2C_0 t).$$

*Proof.* From the definition of $X_t$ in (4), we have $\|X_t\|_2 \le \int_0^t \|b(X_u, u)\|_2 \mathrm{d}u + \|B_t\|_2$. Then, we can get

$$\|X_t\|_2^2 \le 2\left(\int_0^t \|b(X_u, u)\|_2 \mathrm{d}u\right)^2 + 2\|B_t\|_2^2$$

$$\le 2t \int_0^t \|b(X_u, u)\|_2^2 \mathrm{d}u + 2\|B_t\|_2^2$$

$$\le 2t \int_0^t C_0[\|X_u\|_2^2 + 1]\mathrm{d}u + 2\|B_t\|_2^2,$$

where the first inequality holds by the inequality $(a+b)^2 \le 2a^2 + 2b^2$, the second inequality holds by Jensen's inequality, and the last inequality holds by (C1). Thus,

$$\mathbb{E}\|X_t\|_2^2 \le 2t \int_0^t C_0(\mathbb{E}\|X_u\|_2^2 + 1)\mathrm{d}u + 2\mathbb{E}\|B_t\|_2^2$$

$$\le 2C_0 \int_0^t \mathbb{E}\|X_u\|_2^2 \mathrm{d}u + 2(C_0 + p).$$

By Bellman-Gronwall inequality, we have

$$\mathbb{E}\|X_t\|_2^2 \le 2(C_0 + p)\exp(2C_0 t).$$

$\square$

**Lemma A.2.** *Assume (A1) and (A2) hold, then for any $0 \le t_1 \le t_2 \le 1$,*

$$\mathbb{E}[\|X_{t_2} - X_{t_1}\|_2^2] \le 4C_0 \exp(2C_0)(C_0 + p)(t_2 - t_1)^2 + 2C_0(t_2 - t_1)^2 + 2p(t_2 - t_1).$$

*Proof.* From the definition of $X_t$ in (4), we have

$$\|X_{t_2} - X_{t_1}\|_2 \le \int_{t_1}^{t_2} \|b(X_u, u)\|_2 \mathrm{d}u + \|B_{t_2} - B_{t_1}\|_2.$$

Then, we can get

$$\|X_{t_2} - X_{t_1}\|_2^2 \le 2\left(\int_{t_1}^{t_2} \|b(X_u, u)\|_2 \mathrm{d}u\right)^2 + 2\|B_{t_2} - B_{t_1}\|_2^2$$

$$\le 2(t_2 - t_1)\int_{t_1}^{t_2} \|b(X_u, u)\|_2^2 \mathrm{d}u + 2\|B_{t_2} - B_{t_1}\|_2^2$$

$$\le 2(t_2 - t_1)\int_{t_1}^{t_2} C_0[\|X_u\|_2^2 + 1]\mathrm{d}u + 2\|B_{t_2} - B_{t_1}\|_2^2,$$

where the last inequality holds by (C1). Hence,

$$\mathbb{E}\|X_{t_2} - X_{t_1}\|_2^2 \le 2(t_2 - t_1)\int_{t_1}^{t_2} C_0(\mathbb{E}\|X_u\|_2^2 + 1)\mathrm{d}u + 2\mathbb{E}\|B_{t_2} - B_{t_1}\|_2^2$$

$$\le 4C_0 \exp(2C_0)(C_0 + p)(t_2 - t_1)^2 + 2C_0(t_2 - t_1)^2 + 2p(t_2 - t_1),$$

where the last inequality holds by Lemma A.1. $\square$

**Lemma A.3.** *Assume (A1) and (A2) hold, then for any $R > 0$,*

$$\sup_{\|x\|_2 \le R, t \in [0,1]} \mathbb{E}\left[\|b(x,t) - \tilde{b}_m(x,t)\|_2^2\right] \le \mathcal{O}\left(\frac{p \exp(R^2)}{m}\right).$$

*Moreover, if $f$ has a finite upper bound, then*

$$\sup_{x \in \mathbb{R}^p, t \in [0,1]} \mathbb{E}\left[\|b(x,t) - \tilde{b}_m(t,x)\|_2^2\right] \le \mathcal{O}\left(\frac{p}{m}\right).$$

*Proof.* Denote two independent sets of independent copies of $Z \sim N(0, \boldsymbol{I}_p)$, that is, $\mathbf{Z} = \{Z_1, \ldots, Z_m\}$ and $\mathbf{Z}' = \{Z_1', \ldots, Z_m'\}$. For notation convenience, we denote

$$d = \mathbb{E}_Z \nabla f(x + \sqrt{1-t}Z), \; d_m = \frac{\sum_{i=1}^m \nabla f(x + \sqrt{1-t}Z_i)}{m},$$

$$e = \mathbb{E}_Z f(x + \sqrt{1-t}Z), \; e_m = \frac{\sum_{i=1}^m f(x + \sqrt{1-t}Z_i)}{m},$$

$$d_m' = \frac{\sum_{i=1}^m \nabla f(x + \sqrt{1-t}Z_i')}{m}, \; e_m' = \frac{\sum_{i=1}^m f(x + \sqrt{1-t}Z_i')}{m}.$$

Due to $d - d_m = \mathbb{E}\left[d'_m - d_m | \mathbf{Z}\right]$, then $\|d - d_m\|_2^2 \leq \mathbb{E}\left[\|d'_m - d_m\|_2^2 | \mathbf{Z}\right]$. Then,

$$
\begin{aligned}
\mathbb{E}\|d - d_m\|^2 &\leq \mathbb{E}\left[\mathbb{E}[\|d'_m - d_m\|_2^2 | \mathbf{Z}]\right] = \mathbb{E}\|d'_m - d_m\|_2^2 \\
&= \frac{\mathbb{E}_{Z_1, Z'_1}\left\|\nabla f(x + \sqrt{1-t}Z_1) - \nabla f(x + \sqrt{1-t}Z'_1)\right\|_2^2}{m} \\
&\leq \frac{(1-t)\gamma^2}{m}\mathbb{E}_{Z_1, Z'_1}\|Z_1 - Z'_1\|_2^2 \\
&\leq \frac{2p\gamma^2}{m},
\end{aligned}
\tag{8}
$$

where the second inequality holds by (**A1**). Similarly, we also have

$$
\begin{aligned}
\mathbb{E}|e - e_m|^2 &\leq \mathbb{E}|e'_m - e_m|^2 \\
&= \frac{\mathbb{E}_{Z_1, Z'_1}\left|f(x + \sqrt{1-t}Z_1) - f(x + \sqrt{1-t}Z'_1)\right|^2}{m} \\
&\leq \frac{(1-t)\gamma^2}{m}\mathbb{E}_{Z_1, Z'_1}\|Z_1 - Z'_1\|_2^2 \\
&\leq \frac{2p\gamma^2}{m},
\end{aligned}
\tag{9}
$$

where the second inequality holds due to (**A1**). Thus, by (8) and (9), it follows that

$$
\sup_{x \in \mathbb{R}^p, t \in [0,1]} \mathbb{E}\|d - d_m\|_2^2 \leq \frac{2p\gamma^2}{m},
\tag{10}
$$

$$
\sup_{x \in \mathbb{R}^p, t \in [0,1]} \mathbb{E}|e - e_m|^2 \leq \frac{2p\gamma^2}{m}.
\tag{11}
$$

Then, by (**A1**) and (**A2**), through some simple calculation, it follows that

$$
\begin{aligned}
\|b(x,t) - \tilde{b}_m(x,t)\|_2 &= \left\|\frac{d}{e} - \frac{d_m}{e_m}\right\|_2 \\
&\leq \frac{\|d\|_2|e_m - e| + \|d - d_m\|_2|e|}{|ee_m|} \\
&\leq \frac{\gamma|e_m - e| + \|d - d_m\|_2|e|}{\xi^2}.
\end{aligned}
\tag{12}
$$

Let $R > 0$, then

$$
\sup_{\|x\|_2 \leq R} f(x) \leq \mathcal{O}\left(\exp(R^2/2)\right).
\tag{13}
$$

Therefore, by (10)-(13), it can be concluded that

$$
\sup_{\|x\|_2 \leq R, t \in [0,1]} \mathbb{E}\left[\|b(x,t) - \tilde{b}_m(x,t)\|_2^2\right] \leq \mathcal{O}\left(\frac{p\exp(R^2)}{m}\right).
$$

Moreover, if $f$ has a finite upper bound, that is, there exists a positive constant $\zeta$ such that $f \leq \zeta$. Then, similar to (12), it follows that for all $x \in \mathbb{R}^p$ and $t \in [0,1]$,

$$
\|b(x,t) - \tilde{b}_m(x,t)\|_2^2 \leq 2\frac{\gamma^2|e_m - e|^2 + \zeta^2\|d - d_m\|_2^2}{\xi^4}.
\tag{14}
$$

Then, by (10)-(11) and (14), it follows that

$$
\sup_{x \in \mathbb{R}^p, t \in [0,1]} \mathbb{E}\left[\|b(x,t) - \tilde{b}_m(t,x)\|_2^2\right] \leq \mathcal{O}\left(\frac{p}{m}\right).
$$

$\square$

**Lemma A.4.** *Assume (**A1**) and (**A2**) hold, then for $k = 0, 1, \ldots, K$,*

$$E[\|\widetilde{Y}_{t_k}\|_2^2] \leq \frac{6\gamma^2}{\xi^2} + 3p.$$

*Proof.* Define $\Theta_{k,t} = \widetilde{Y}_{t_k} + (t - t_k)\tilde{b}_m(\widetilde{Y}_{t_k}, t_k)$ and $\widetilde{Y}_t = \Theta_{k,t} + B_t - B_{t_k}$, where $t_k \leq t \leq t_{k+1}$ with $k = 0, 1, \ldots, K - 1$. By (**A1**) and (**A2**), it follows that for all $x \in \mathbb{R}^p$ and $t \in [0, 1]$,

$$\|b(x, t)\|_2^2 \leq \frac{\gamma^2}{\xi^2}, \quad \|\tilde{b}_m(x, t)\|_2^2 \leq \frac{\gamma^2}{\xi^2}. \tag{15}$$

Then, by (15), we have

$$\|\Theta_{k,t}\|_2^2 = \|\widetilde{Y}_{t_k}\|_2^2 + (t - t_k)^2\|\tilde{b}_m(\widetilde{Y}_{t_k}, t_k)\|_2^2 + 2(t - t_k)\widetilde{Y}_{t_k}^\top \tilde{b}_m(\widetilde{Y}_{t_k}, t_k)$$
$$\leq (1 + s)\|\widetilde{Y}_{t_k}\|_2^2 + \frac{(s + s^2)\gamma^2}{\xi^2},$$

where the inequality by using $(a + b)^2 \leq 2a^2 + 2b^2$. Further, we can get

$$\mathbb{E}[\|\widetilde{Y}_t\|_2^2|\widetilde{Y}_{t_k}] = \mathbb{E}[\|\Theta_{k,t}\|_2^2|\widetilde{Y}_{t_k}] + (t - t_k)p$$
$$\leq (1 + s)\mathbb{E}\|\widetilde{Y}_{t_k}\|_2^2 + \frac{(s + s^2)\gamma^2}{\xi^2} + sp.$$

Therefore,

$$\mathbb{E}[\|\widetilde{Y}_{t_{k+1}}\|_2^2] \leq (1 + s)\mathbb{E}\|\widetilde{Y}_{t_k}\|_2^2 + \frac{(s + s^2)\gamma^2}{\xi^2} + sp.$$

Since $\widetilde{Y}_{t_0} = 0$, then by induction, we have

$$\mathbb{E}[\|\widetilde{Y}_{t_{k+1}}\|_2^2] \leq (1 + s)^2\mathbb{E}[\|\widetilde{Y}_{t_{k-1}}\|_2^2] + (1 + s)\left[\frac{(s + s^2)\gamma^2}{\xi^2} + sp\right] + \frac{(s + s^2)\gamma^2}{\xi^2} + sp$$

$$\ldots\ldots$$

$$\leq ((1 + s)^k + (1 + s)^{k-1} + \ldots + 1)\left[\frac{(s + s^2)\gamma^2}{\xi^2} + sp\right]$$

$$\leq \frac{6\gamma^2}{\xi^2} + 3p.$$

$\square$

**Lemma A.5.** *Assume (**A1**) and (**A2**) hold, then for $k = 0, 1, \ldots, K$,*

$$\mathbb{E}\left\|b(\widetilde{Y}_{t_k}, t_k) - \tilde{b}_m(\widetilde{Y}_{t_k}, t_k)\right\|_2^2 \leq \mathcal{O}\left(\frac{p}{\log(m)}\right).$$

*Moreover, if $f$ has a finite upper bound, then*

$$\mathbb{E}\left\|b(\widetilde{Y}_{t_k}, t_k) - \tilde{b}_m(\widetilde{Y}_{t_k}, t_k)\right\|_2^2 \leq \mathcal{O}\left(\frac{p}{m}\right).$$

*Proof.* Let $R > 0$, then

$$\mathbb{E}\left\|b(\widetilde{Y}_{t_k}, t_k) - \tilde{b}_m(\widetilde{Y}_{t_k}, t_k)\right\|_2^2 = \mathbb{E}_{\widetilde{Y}_{t_k}}\mathbb{E}_Z\left[\left\|b(\widetilde{Y}_{t_k}, t_k) - \tilde{b}_m(\widetilde{Y}_{t_k}, t_k)\right\|_2^2 1(\|\widetilde{Y}_{t_k}\|_2 \leq R)\right]$$
$$+ \mathbb{E}_{\widetilde{Y}_{t_k}}\mathbb{E}_Z\left[\left\|b(\widetilde{Y}_{t_k}, t_k) - \tilde{b}_m(\widetilde{Y}_{t_k}, t_k)\right\|_2^2 1(\|\widetilde{Y}_{t_k}\|_2 > R)\right]. \tag{16}$$

Next, we need to bound the two terms of (16). First, by Lemma A.3, we have

$$\mathbb{E}_{\widetilde{Y}_{t_k}} \mathbb{E}_Z \left[ \left\| b(\widetilde{Y}_{t_k}, t_k) - \tilde{b}_m(\widetilde{Y}_{t_k}, t_k) \right\|_2^2 \mathbb{1}(\|\widetilde{Y}_{t_k}\|_2 \leq R) \right] \leq \mathcal{O}\left( \frac{p \exp(R^2)}{m} \right).$$

Secondly, combining (15) and Lemma A.4 with Markov inequality, it follows that

$$\mathbb{E}_{\widetilde{Y}_{t_k}} \mathbb{E}_Z \left[ \left\| b(\widetilde{Y}_{t_k}, t_k) - \tilde{b}_m(\widetilde{Y}_{t_k}, t_k) \right\|_2^2 \mathbb{1}(\|\widetilde{Y}_{t_k}\|_2 > R) \right] \leq \mathcal{O}\left( p/R^2 \right).$$

Thence

$$\mathbb{E} \left\| b(\widetilde{Y}_{t_k}, t_k) - \tilde{b}_m(\widetilde{Y}_{t_k}, t_k) \right\|_2^2 \leq \mathcal{O}\left( \frac{p \exp(R^2)}{m} \right) + \mathcal{O}\left( p/R^2 \right). \tag{17}$$

Set $R = \left( \frac{\log(m)}{2} \right)^{1/2}$ in (17), then we have

$$\mathbb{E} \left\| b(\widetilde{Y}_{t_k}, t_k) - \tilde{b}_m(\widetilde{Y}_{t_k}, t_k) \right\|_2^2 \leq \mathcal{O}\left( \frac{p}{\log(m)} \right).$$

Moreover, if $f$ has a finite upper bound, then by Lemma A.3, we can similarly get

$$\mathbb{E} \left\| b(\widetilde{Y}_{t_k}, t_k) - \tilde{b}_m(\widetilde{Y}_{t_k}, t_k) \right\|_2^2 = \mathbb{E}_{\widetilde{Y}_{t_k}} \mathbb{E}_Z \left[ \left\| b(\widetilde{Y}_{t_k}, t_k) - \tilde{b}_m(\widetilde{Y}_{t_k}, t_k) \right\|_2^2 \right] \leq \mathcal{O}\left( \frac{p}{m} \right).$$

This completes the proof. $\qquad \square$

### A.3 Proof of Theorem 3.1

*Proof.* From the definition of $\widetilde{Y}_{t_k}$ and $X_{t_k}$, we have

$$\|\widetilde{Y}_{t_k} - X_{t_k}\|_2^2$$

$$\leq \|\widetilde{Y}_{t_{k-1}} - X_{t_{k-1}}\|_2^2 + \left( \int_{t_{k-1}}^{t_k} \|b(X_u, u) - \tilde{b}_m(\widetilde{Y}_{t_{k-1}}, t_{k-1})\|_2 \mathrm{d}u \right)^2$$

$$+ 2\|\widetilde{Y}_{t_{k-1}} - X_{t_{k-1}}\|_2 \left( \int_{t_{k-1}}^{t_k} \|b(X_u, u) - \tilde{b}_m(\widetilde{Y}_{t_{k-1}}, t_{k-1})\|_2 \mathrm{d}u \right)$$

$$\leq (1+s)\|\widetilde{Y}_{t_{k-1}} - X_{t_{k-1}}\|_2^2 + (1+s) \int_{t_{k-1}}^{t_k} \|b(X_u, u) - \tilde{b}_m(\widetilde{Y}_{t_{k-1}}, t_{k-1})\|_2^2 \mathrm{d}u$$

$$\leq (1+s)\|\widetilde{Y}_{t_{k-1}} - X_{t_{k-1}}\|_2^2 + 2(1+s) \int_{t_{k-1}}^{t_k} \|b(X_u, u) - b(\widetilde{Y}_{t_{k-1}}, t_{k-1})\|_2^2 \mathrm{d}u$$

$$+ 2s(1+s)\|b(\widetilde{Y}_{t_{k-1}}, t_{k-1}) - \tilde{b}_m(\widetilde{Y}_{t_{k-1}}, t_{k-1})\|_2^2$$

$$\leq (1+s)\|\widetilde{Y}_{t_{k-1}} - X_{t_{k-1}}\|_2^2 + 4C_1^2(1+s) \int_{t_{k-1}}^{t_k} [\|X_u - \widetilde{Y}_{t_{k-1}}\|_2^2 + |u - t_{k-1}|] \mathrm{d}u$$

$$+ 2s(1+s)\|b(\widetilde{Y}_{t_{k-1}}, t_{k-1}) - \tilde{b}_m(\widetilde{Y}_{t_{k-1}}, t_{k-1})\|_2^2$$

$$\leq (1+s)\|\widetilde{Y}_{t_{k-1}} - X_{t_{k-1}}\|_2^2 + 8C_1^2(1+s) \int_{t_{k-1}}^{t_k} \|X_u - X_{t_{k-1}}\|_2^2 \mathrm{d}u$$

$$+ 8C_1^2 s(1+s)\|X_{t_{k-1}} - \widetilde{Y}_{t_{k-1}}\|_2^2 + 4C_1^2(1+s)s^2$$

$$+ 2s(1+s)\|b(\widetilde{Y}_{t_{k-1}}, t_{k-1}) - \tilde{b}_m(\widetilde{Y}_{t_{k-1}}, t_{k-1})\|_2^2$$

$$\leq (1+s+8C_1^2(s+s^2))\|\widetilde{Y}_{t_{k-1}} - X_{t_{k-1}}\|_2^2 + 8C_1^2(1+s) \int_{t_{k-1}}^{t_k} \|X_u - X_{t_{k-1}}\|_2^2 \mathrm{d}u$$

$$+ 4C_1^2(1+s)s^2 + 2s(1+s)\|b(\widetilde{Y}_{t_{k-1}}, t_{k-1}) - \tilde{b}_m(\widetilde{Y}_{t_{k-1}}, t_{k-1})\|_2^2,$$

where the second inequality holds due to $2ab \leq sa^2 + \frac{b^2}{s}$ with $a = \|\widetilde{Y}_{t_{k-1}} - X_{t_{k-1}}\|_2$ and $b = \int_{t_{k-1}}^{t_k} \|b(X_u, u) - \tilde{b}_m(\widetilde{Y}_{t_{k-1}}, t_{k-1})\|_2 du$ and $b^2 \leq s \cdot \int_{t_{k-1}}^{t_k} \|b(X_u, u) - \tilde{b}_m(\widetilde{Y}_{t_{k-1}}, t_{k-1})\|_2^2 du$ , the fourth inequality holds by (C3). Then,

$$
\begin{aligned}
\mathbb{E}\|\widetilde{Y}_{t_k} - X_{t_k}\|_2^2 &\leq (1 + s + 8C_1^2(s + s^2))\mathbb{E}\|\widetilde{Y}_{t_{k-1}} - X_{t_{k-1}}\|_2^2 \\
&\quad + 8C_1^2(1+s)\int_{t_{k-1}}^{t_k} \mathbb{E}\|X_u - X_{t_{k-1}}\|_2^2 du + 4C_1^2(s^2 + s^3) \\
&\quad + 2s(1+s)\mathbb{E}[\|b(\widetilde{Y}_{t_{k-1}}, t_{k-1}) - \tilde{b}_m(\widetilde{Y}_{t_{k-1}}, t_{k-1})\|_2^2] \\
&\leq (1 + s + 8C_1^2(s + s^2))\mathbb{E}\|\widetilde{Y}_{t_{k-1}} - X_{t_{k-1}}\|_2^2 + h(s) \\
&\quad + 4C_1^2(s^2 + s^3) + 2s(1+s)\mathbb{E}[\|b(\widetilde{Y}_{t_{k-1}}, t_{k-1}) - \tilde{b}_m(\widetilde{Y}_{t_{k-1}}, t_{k-1})\|_2^2] \\
&\leq (1 + s + 8C_1^2(s + s^2))\mathbb{E}\|\widetilde{Y}_{t_{k-1}} - X_{t_{k-1}}\|_2^2 + h(s) \\
&\quad + 4C_1^2(s^2 + s^3) + 2s(1+s)\mathcal{O}\left(\frac{p}{\log(m)}\right),
\end{aligned}
\tag{18}
$$

where $h(s) = 8C_1^2(s + s^2)[4C_0 \exp(2C_0)(C_0 + p)s^2 + 2C_0 s^2 + 2ps]$, and the last inequality holds by Lemma A.5. Owing to $\widetilde{Y}_{t_0} = X_{t_0} = 0$, we can conclude that

$$
\begin{aligned}
&\mathbb{E}\|\widetilde{Y}_{t_K} - X_{t_K}\|_2^2 \\
&\leq \frac{(1 + s + 8C_1^2(s + s^2))^K - 1}{s + 8C_1^2(s + s^2)}\left[h(s) + 4C_1^2(s^2 + s^3) + 2(s + s^2)\mathcal{O}\left(\frac{p}{\log(m)}\right)\right] \\
&\leq \mathcal{O}(ps) + \mathcal{O}\left(\frac{p}{\log(m)}\right).
\end{aligned}
$$

Therefore,

$$
W_2(Law(\widetilde{Y}_{t_K}), \mu) \leq \mathcal{O}(\sqrt{ps}) + \mathcal{O}\left(\sqrt{\frac{p}{\log(m)}}\right).
$$

$\square$

## A.4 Proof of Theorem 3.2

*Proof.* This proof is same as that of Theorem 3.1. Similar to (18), by Lemma A.5, it follows that

$$
\begin{aligned}
\mathbb{E}\|\widetilde{Y}_{t_k} - X_{t_k}\|_2^2 &\leq (1 + s + 8C_1^2(s + s^2))\mathbb{E}\|\widetilde{Y}_{t_{k-1}} - X_{t_{k-1}}\|_2^2 + h(s) \\
&\quad + 4C_1^2(s^2 + s^3) + 2s(1+s)\mathcal{O}\left(\frac{p}{m}\right),
\end{aligned}
$$

where $s = \frac{1}{K}$ is the step size and $t_k = ks$. Then, we also have

$$
\begin{aligned}
&\mathbb{E}\|\widetilde{Y}_{t_K} - X_{t_K}\|_2^2 \\
&\leq \frac{(1 + s + 8C_1^2(s + s^2))^K - 1}{s + 8C_1^2(s + s^2)}\left[h(s) + 4C_1^2(s^2 + s^3) + 2(s + s^2)\mathcal{O}\left(\frac{1}{m}\right)\right] \\
&\leq \mathcal{O}(ps) + \mathcal{O}\left(\frac{p}{m}\right).
\end{aligned}
$$

Hence, it follows that

$$
W_2(Law(\widetilde{Y}_{t_K}), \mu) \leq \mathcal{O}(\sqrt{ps}) + \mathcal{O}\left(\sqrt{\frac{p}{m}}\right).
$$

$\square$

### A.5 Preliminary lemmas for Theorem 3.3

To prove Theorem 3.3, we first prove the Lemmas A.6-A.8.

**Lemma A.6.** *Assume (**A1**) holds, then for any $R > 0$,*

$$\sup_{\|x\|_2 \le R, t \in [0,1]} \mathbb{E}\left[\|b(x,t) - \tilde{b}_m(x,t)\|_2^2\right] \le \mathcal{O}\left(\frac{p\exp(R^2)(C_p)^4}{m\varepsilon^4}\right) + \mathcal{O}\left(\frac{p(C_p)^2}{m\varepsilon^2}\right),$$

*where $C_p = (2\pi)^{p/2}C^{-1}$. Moreover, if $f$ has a finite upper bound, then*

$$\sup_{x \in \mathbb{R}^p, t \in [0,1]} \mathbb{E}\left[\|b(x,t) - \tilde{b}_m(t,x)\|_2^2\right] \le \mathcal{O}\left(\frac{p(C_p)^4}{m\varepsilon^4}\right) + \mathcal{O}\left(\frac{p(C_p)^2}{m\varepsilon^2}\right).$$

*Proof.* Denote two independent sets of independent copies of $Z \sim N(0, \boldsymbol{I}_p)$ by $\mathbf{Z} = \{Z_1, \ldots, Z_m\}$ and $\mathbf{Z}' = \{Z_1', \ldots, Z_m'\}$. For notation convenience, we denote

$$d = \mathbb{E}_Z \nabla g(x + \sqrt{1-t}Z), \ d_m = \frac{\sum_{i=1}^m \nabla g(x + \sqrt{1-t}Z_i)}{m},$$

$$e = \mathbb{E}_Z \left[g(x + \sqrt{1-t}Z) + \frac{\varepsilon}{C_p(1-\varepsilon)}\right], \ e_m = \frac{\sum_{i=1}^m g(x + \sqrt{1-t}Z_i)}{m} + \frac{\varepsilon}{C_p(1-\varepsilon)},$$

$$d_m' = \frac{\sum_{i=1}^m \nabla g(x + \sqrt{1-t}Z_i')}{m}, \ e_m' = \frac{\sum_{i=1}^m g(x + \sqrt{1-t}Z_i')}{m} + \frac{\varepsilon}{C_p(1-\varepsilon)},$$

where $g(x) = \exp(\|x\|_2^2/2 - V(x))$. Since $d - d_m = \mathbb{E}\left[d_m' - d_m|\mathbf{Z}\right]$, we have $\|d - d_m\|_2^2 \le \mathbb{E}\left[\|d_m' - d_m\|_2^2|\mathbf{Z}\right]$. By (**A1**), it follows that $g$ and $\nabla g$ are Lipschitz continuous. Thus there exists a positive constant $\gamma$ such that for all $x, y \in \mathbb{R}^p$,

$$|g(x) - g(y)| \le \gamma\|x - y\|_2, \tag{19}$$

$$\|\nabla g(x) - \nabla g(y)\|_2 \le \gamma\|x - y\|_2. \tag{20}$$

Therefore,

$$\begin{aligned}
\mathbb{E}\|d - d_m\|^2 &\le \mathbb{E}\left[\mathbb{E}[\|d_m' - d_m\|_2^2|\mathbf{Z}]\right] = \mathbb{E}\|d_m' - d_m\|_2^2 \\
&= \frac{\mathbb{E}_{Z_1, Z_1'}\left\|\nabla g(x + \sqrt{1-t}Z_1) - \nabla g(x + \sqrt{1-t}Z_1')\right\|_2^2}{m} \\
&\le \frac{(1-t)\gamma^2}{m}\mathbb{E}_{Z_1, Z_1'}\|Z_1 - Z_1'\|_2^2 \\
&\le \frac{2p\gamma^2}{m},
\end{aligned} \tag{21}$$

where the second inequality follows from (20). Similarly, we also have

$$\begin{aligned}
\mathbb{E}|e - e_m|^2 &\le \mathbb{E}|e_m' - e_m|^2 \\
&= \frac{\mathbb{E}_{Z_1, Z_1'}\left|g(x + \sqrt{1-t}Z_1) - g(x + \sqrt{1-t}Z_1')\right|^2}{m} \\
&\le \frac{(1-t)\gamma^2}{m}\mathbb{E}_{Z_1, Z_1'}\|Z_1 - Z_1'\|_2^2 \\
&\le \frac{2p\gamma^2}{m},
\end{aligned} \tag{22}$$

where the second inequality follows from (19).Hence, by (21) and (22), we have

$$\sup_{x \in \mathbb{R}^p, t \in [0,1]} \mathbb{E}\|d - d_m\|_2^2 \le \frac{2p\gamma^2}{m}, \tag{23}$$

$$\sup_{x \in \mathbb{R}^p, t \in [0,1]} \mathbb{E}|e - e_m|^2 \leq \frac{2p\gamma^2}{m}. \tag{24}$$

Then, by (19) and (20), through some simple calculation, it follows that

$$
\begin{aligned}
\|b(x,t) - \tilde{b}_m(x,t)\|_2 &= \left\| \frac{d}{e} - \frac{d_m}{e_m} \right\|_2 \\
&\leq \frac{\|d\|_2 |e_m - e| + \|d - d_m\|_2 |e|}{|e e_m|} \\
&\leq \frac{\gamma |e_m - e| + \|d - d_m\|_2 |e|}{(\varepsilon/(C_p - C_p \varepsilon))^2}.
\end{aligned} \tag{25}
$$

Let $R > 0$, then

$$\sup_{\|x\|_2 \leq R} g(x) \leq \mathcal{O}\left(\exp(R^2/2)\right). \tag{26}$$

Therefore, by (23)-(26), it can be concluded that

$$
\begin{aligned}
\sup_{\|x\|_2 \leq R, t \in [0,1]} \mathbb{E}\left[\|b(x,t) - \tilde{b}_m(x,t)\|_2^2\right] &\leq \mathcal{O}\left(\frac{p \exp(R^2)}{m(\varepsilon/(C_p - C_p \varepsilon))^4}\right) + \mathcal{O}\left(\frac{p}{m(\varepsilon/(C_p - C_p \varepsilon))^2}\right) \\
&\leq \mathcal{O}\left(\frac{p \exp(R^2)(C_p)^4}{m \varepsilon^4}\right) + \mathcal{O}\left(\frac{p(C_p)^2}{m \varepsilon^2}\right).
\end{aligned}
$$

Moreover, $f$ has a finite upper bound so does $g$. Then there exists a positive constant $\zeta$ such that $g \leq \zeta$. Similar to (25), it follows that for all $x \in \mathbb{R}^p$ and $t \in [0,1]$,

$$\|b(x,t) - \tilde{b}_m(x,t)\|_2^2 \leq 2\frac{\gamma^2 |e_m - e|^2 + (\zeta + \varepsilon/(C_p - C_p \varepsilon))^2 \|d - d_m\|_2^2}{(\varepsilon/(C_p - C_p \varepsilon))^4}. \tag{27}$$

Then, by (23)-(24) and (27), it follows that

$$
\begin{aligned}
\sup_{x \in \mathbb{R}^p, t \in [0,1]} \mathbb{E}\left[\|b(x,t) - \tilde{b}_m(t,x)\|_2^2\right] &\leq \mathcal{O}\left(\frac{p}{m(\varepsilon/(C_p - C_p \varepsilon))^4}\right) + \mathcal{O}\left(\frac{p}{m(\varepsilon/(C_p - C_p \varepsilon))^2}\right) \\
&\leq \mathcal{O}\left(\frac{p(C_p)^4}{m \varepsilon^4}\right) + \mathcal{O}\left(\frac{p(C_p)^2}{m \varepsilon^2}\right).
\end{aligned}
$$

$\square$

**Lemma A.7.** *Assume (**A1**) holds, then for $k = 0, 1, \ldots, K$,*

$$\mathbb{E}[\|\widetilde{Y}_{t_k}^\varepsilon\|_2^2] \leq \mathcal{O}\left(\frac{(C_p)^2}{\varepsilon^2}\right) + \mathcal{O}(p),$$

*where $C_p = (2\pi)^{p/2} C^{-1}$.*

*Proof.* Define $\Theta_{k,t}^\varepsilon = \widetilde{Y}_{t_k}^\varepsilon + (t - t_k)\tilde{b}_m(\widetilde{Y}_{t_k}^\varepsilon, t_k)$ and $\widetilde{Y}_t^\varepsilon = \Theta_{k,t}^\varepsilon + B_t - B_{t_k}$, where $t_k \leq t \leq t_{k+1}$ with $k = 0, 1, \ldots, K - 1$. By (**A1**), then there exists a positive constant $\gamma$ such that $g$ is $\gamma$-Lipschitz continuous. Then, for all $x \in \mathbb{R}^p$ and $t \in [0,1]$, we have

$$\|b(x,t)\|_2^2 \leq \frac{\gamma^2}{(\varepsilon/(C_p - C_p \varepsilon))^2}, \quad \|\tilde{b}_m(x,t)\|_2^2 \leq \frac{\gamma^2}{(\varepsilon/(C_p - C_p \varepsilon))^2}. \tag{28}$$

By (28), we have

$$
\begin{aligned}
\|\Theta_{k,t}^\varepsilon\|_2^2 &= \|\widetilde{Y}_{t_k}^\varepsilon\|_2^2 + (t - t_k)^2 \|\tilde{b}_m(\widetilde{Y}_{t_k}^\varepsilon, t_k)\|_2^2 + 2(t - t_k)(\widetilde{Y}_{t_k}^\varepsilon)^\top \tilde{b}_m(\widetilde{Y}_{t_k}^\varepsilon, t_k) \\
&\leq (1 + s)\|\widetilde{Y}_{t_k}^\varepsilon\|_2^2 + \frac{(s + s^2)\gamma^2}{(\varepsilon/(C_p - C_p \varepsilon))^2}.
\end{aligned}
$$

Furthermore, it can be shown that

$$\mathbb{E}[\|\widetilde{Y}_t^\varepsilon\|_2^2|\widetilde{Y}_{t_k}^\varepsilon] = \mathbb{E}[\|\Theta_{k,t}^\varepsilon\|_2^2|\widetilde{Y}_{t_k}^\varepsilon] + (t-t_k)p$$

$$\leq (1+s)\mathbb{E}\|\widetilde{Y}_{t_k}^\varepsilon\|_2^2 + \frac{(s+s^2)\gamma^2}{(\varepsilon/(C_p - C_p\varepsilon))^2} + sp.$$

Therefore,

$$\mathbb{E}[\|\widetilde{Y}_{t_{k+1}}^\varepsilon\|_2^2] \leq (1+s)\mathbb{E}[\|\widetilde{Y}_{t_k}^\varepsilon\|_2^2] + \frac{(s+s^2)\gamma^2}{(\varepsilon/(C_p - C_p\varepsilon))^2} + sp.$$

Since $\widetilde{Y}_{t_0}^\varepsilon = 0$, then by induction, we have

$$\mathbb{E}[\|\widetilde{Y}_{t_{k+1}}^\varepsilon\|_2^2] \leq \frac{6\gamma^2}{(\varepsilon/(C_p - C_p\varepsilon))^2} + 3p \leq \mathcal{O}\left(\frac{(C_p)^2}{\varepsilon^2}\right) + \mathcal{O}\left(p\right).$$

$\square$

**Lemma A.8.** *Assume (**A1**) holds, then for $k = 0, 1, \ldots, K$ and $t \in [0,1]$,*

$$\mathbb{E}\left\|b(\widetilde{Y}_{t_k}^\varepsilon, t_k) - \tilde{b}_m(\widetilde{Y}_{t_k}^\varepsilon, t_k)\right\|_2^2 \leq \mathcal{O}\left(\frac{p(C_p)^4}{\sqrt{m}\varepsilon^4}\right) + \mathcal{O}\left(\frac{(C_p)^4}{\log(m)\varepsilon^4}\right) + \mathcal{O}\left(\frac{p(C_p)^2}{\log(m)\varepsilon^2}\right),$$

*where $C_p = (2\pi)^{p/2}C^{-1}$. Moreover, if $f$ has a finite upper bound, then*

$$\mathbb{E}\left\|b(\widetilde{Y}_{t_k}^\varepsilon, t_k) - \tilde{b}_m(\widetilde{Y}_{t_k}^\varepsilon, t_k)\right\|_2^2 \leq \mathcal{O}\left(\frac{p(C_p)^4}{m\varepsilon^4}\right) + \mathcal{O}\left(\frac{p(C_p)^2}{m\varepsilon^2}\right).$$

*Proof.* Let $R > 0$, then

$$\mathbb{E}\left\|b(\widetilde{Y}_{t_k}^\varepsilon, t_k) - \tilde{b}_m(\widetilde{Y}_{t_k}^\varepsilon, t_k)\right\|_2^2 = \mathbb{E}_{\widetilde{Y}_{t_k}^\varepsilon} \mathbb{E}_Z \left[\left\|b(\widetilde{Y}_{t_k}^\varepsilon, t_k) - \tilde{b}_m(\widetilde{Y}_{t_k}^\varepsilon, t_k)\right\|_2^2 \mathbb{1}(\|\widetilde{Y}_{t_k}^\varepsilon\|_2 \leq R)\right]$$
$$+ \mathbb{E}_{\widetilde{Y}_{t_k}^\varepsilon} \mathbb{E}_Z \left[\left\|b(\widetilde{Y}_{t_k}^\varepsilon, t_k) - \tilde{b}_m(\widetilde{Y}_{t_k}^\varepsilon, t_k)\right\|_2^2 \mathbb{1}(\|\widetilde{Y}_{t_k}^\varepsilon\|_2 > R)\right]. \tag{29}$$

Next, we need to bound the two terms on the right hand of (29). First, by Lemma A.6, we have

$$\mathbb{E}_{\widetilde{Y}_{t_k}^\varepsilon} \mathbb{E}_Z \left[\left\|b(\widetilde{Y}_{t_k}^\varepsilon, t_k) - \tilde{b}_m(\widetilde{Y}_{t_k}^\varepsilon, t_k)\right\|_2^2 \mathbb{1}(\|\widetilde{Y}_{t_k}^\varepsilon\|_2 \leq R)\right] \leq \mathcal{O}\left(\frac{p\exp(R^2)(C_p)^4}{m\varepsilon^4}\right) + \mathcal{O}\left(\frac{p(C_p)^2}{m\varepsilon^2}\right).$$

Second, by combining (28) and Lemma A.7 with the Markov inequality, we have

$$\mathbb{E}_{\widetilde{Y}_{t_k}^\varepsilon} \mathbb{E}_Z \left[\left\|b(\widetilde{Y}_{t_k}^\varepsilon, t_k) - \tilde{b}_m(\widetilde{Y}_{t_k}^\varepsilon, t_k)\right\|_2^2 \mathbb{1}(\|\widetilde{Y}_{t_k}^\varepsilon\|_2 > R)\right] \leq \mathcal{O}\left(\frac{(C_p)^4}{R^2\varepsilon^4}\right) + \mathcal{O}\left(\frac{p(C_p)^2}{R^2\varepsilon^2}\right).$$

Therefore,

$$\mathbb{E}\left\|b(\widetilde{Y}_{t_k}^\varepsilon, t_k) - \tilde{b}_m(\widetilde{Y}_{t_k}^\varepsilon, t_k)\right\|_2^2 \leq \mathcal{O}\left(\frac{p\exp(R^2)(C_p)^4}{m\varepsilon^4}\right) + \mathcal{O}\left(\frac{p(C_p)^2}{m\varepsilon^2}\right)$$
$$+ \mathcal{O}\left(\frac{(C_p)^4}{R^2\varepsilon^4}\right) + \mathcal{O}\left(\frac{p(C_p)^2}{R^2\varepsilon^2}\right). \tag{30}$$

Setting $R = \left(\frac{\log(m)}{2}\right)^{1/2}$ in (30), we have

$$\mathbb{E}\left\|b(\widetilde{Y}_{t_k}^\varepsilon, t_k) - \tilde{b}_m(\widetilde{Y}_{t_k}^\varepsilon, t_k)\right\|_2^2 \leq \mathcal{O}\left(\frac{p(C_p)^4}{\sqrt{m}\varepsilon^4}\right) + \mathcal{O}\left(\frac{(C_p)^4}{\log(m)\varepsilon^4}\right) + \mathcal{O}\left(\frac{p(C_p)^2}{\log(m)\varepsilon^2}\right).$$

Moreover, if $f$ has a finite upper bound, then by Lemma A.6, we can similarly get

$$\mathbb{E}\left\|b(\widetilde{Y}_{t_k}^\varepsilon, t_k) - \tilde{b}_m(\widetilde{Y}_{t_k}^\varepsilon, t_k)\right\|_2^2 = \mathbb{E}_{\widetilde{Y}_{t_k}^\varepsilon}\mathbb{E}_Z\left[\left\|b(\widetilde{Y}_{t_k}^\varepsilon, t_k) - \tilde{b}_m(\widetilde{Y}_{t_k}^\varepsilon, t_k)\right\|_2^2\right]$$
$$\leq \mathcal{O}\left(\frac{p(C_p)^4}{m\varepsilon^4}\right) + \mathcal{O}\left(\frac{p(C_p)^2}{m\varepsilon^2}\right).$$

This completes the proof. $\qquad\square$

## A.6  Proof of Theorem 3.3

*Proof.* By triangle inequality, we have $W_2(Law(\widetilde{Y}_{t_K}^\varepsilon), \mu) \leq W_2(Law(\widetilde{Y}_{t_K}^\varepsilon), \mu_\varepsilon) + W_2(\mu, \mu_\varepsilon)$, then we obtain the upper bound of two terms on the right hand of this inequality, respectively.

First, similar to the proof of Theorem 3.1, by Lemma A.8 and $\widetilde{Y}_{t_0}^\varepsilon = X_{t_0} = 0$ and through some calculation, we can conclude that

$$W_2(Law(\widetilde{Y}_{t_K}^\varepsilon), \mu_\varepsilon) \leq \mathcal{O}(\sqrt{ps}) + \mathcal{O}\left(\frac{\sqrt{p}(C_p)^2}{m^{1/4}\varepsilon^2}\right) + \mathcal{O}\left(\frac{(C_p)^2}{\sqrt{\log(m)}\varepsilon^2}\right) + \mathcal{O}\left(\frac{\sqrt{p}C_p}{\sqrt{\log(m)}\varepsilon}\right). \tag{31}$$

Second, we need to get the upper bound of $W_2(\mu, \mu_\varepsilon)$. Let $Y \sim \mu$ and $Z \sim N(0, \boldsymbol{I}_p)$, $\theta$ is one Bernoulli random variable satisfying $P(\theta = 1) = 1 - \varepsilon$ and $P(\theta = 0) = \varepsilon$. Assume $Y$, $Z$ and $\theta$ are independent of each other. Then $(Y, (1 - \theta)Z + \theta Y)$ is one coupling of $(\mu, \mu_\varepsilon)$, and denote its joint distribution by $\pi$. Therefore, we have

$$\int_{\mathbb{R}^p \times \mathbb{R}^p} \|x - y\|_2^2 d\pi = \mathbb{E}\|Y - ((1 - \theta)Z + \theta Y)\|_2^2$$
$$= \mathbb{E}\left[\mathbb{E}\left[\|Y - ((1 - \theta)Z + \theta Y)\|_2^2 | \theta\right]\right]$$
$$= \mathbb{E}\left[\mathbb{E}\left[\|Y - ((1 - \theta)Z + \theta Y)\|_2^2 | \theta = 1\right]\right] P(\theta = 1)$$
$$\quad + \mathbb{E}\left[\mathbb{E}\left[\|Y - ((1 - \theta)Z + \theta Y)\|_2^2 | \theta = 0\right]\right] P(\theta = 0)$$
$$= \mathbb{E}[\|Y - Z\|_2^2 | \theta = 0] P(\theta = 0)$$
$$= \varepsilon\mathbb{E}\|Y - Z\|_2^2$$
$$\leq \mathcal{O}(p\varepsilon).$$

Then we have

$$W_2(\mu, \mu_\varepsilon) \leq \mathcal{O}(\sqrt{p\varepsilon}). \tag{32}$$

Combining (31) with (32), it follows that

$$W_2(Law(\widetilde{Y}_{t_K}^\varepsilon), \mu)$$
$$\leq \mathcal{O}(\sqrt{p\varepsilon}) + \mathcal{O}(\sqrt{ps}) + \mathcal{O}\left(\frac{\sqrt{p}(C_p)^2}{m^{1/4}\varepsilon^2}\right) + \mathcal{O}\left(\frac{(C_p)^2}{\sqrt{\log(m)}\varepsilon^2}\right) + \mathcal{O}\left(\frac{\sqrt{p}C_p}{\sqrt{\log(m)}\varepsilon}\right). \tag{33}$$

Set $\varepsilon = (\log(m))^{-1/5}$ in (33), then there exist one constant $\widetilde{C}_p$ depending on $p$ such that

$$W_2(Law(\widetilde{Y}_{t_K}^\varepsilon), \mu) \leq \widetilde{C}_p \cdot \mathcal{O}\left(\frac{1}{(\log(m))^{1/10}}\right) + \mathcal{O}(\sqrt{ps}).$$

Moreover, if $f$ has a finite upper bound, then similar to the proof of Theorem 3.2 and by (32) and Lemma A.8, we have

$$W_2(Law(\widetilde{Y}_{t_K}^\varepsilon), \mu) \leq \mathcal{O}(\sqrt{p\varepsilon}) + \mathcal{O}(\sqrt{ps}) + \mathcal{O}\left(\frac{\sqrt{p}(C_p)^2}{\sqrt{m}\varepsilon^2}\right) + \mathcal{O}\left(\frac{\sqrt{p}C_p}{\sqrt{m}\varepsilon}\right). \tag{34}$$

Set $\varepsilon = m^{-1/5}$ in (34), then there exists one constant $\widetilde{C}_p$ depending on $p$ such that

$$W_2(Law(\widetilde{Y}^\varepsilon_{t_K}), \mu) \leq \widetilde{C}_p \cdot \mathcal{O}\left(\frac{1}{m^{1/10}}\right) + \mathcal{O}(\sqrt{ps}).$$

$\square$

