# OpenReview forum: "Convergence Analysis of Schr{\"o}dinger-F{\"o}llmer Sampler without Convexity"
_TMLR — Withdrawn by Authors_

### Review · Reviewer_yQB8 · 2022-06-10

**Summary Of Contributions:**

This paper deals with the theoretical analysis of the convergence of the Schrodinger-Follmer sampler, an alternative to Markov Chain Monte Carlo algorithms for sampling from an unnormalized density. The authors start by recalling the methodology of the Schrodinger-Follmer sampler and recall the results obtain in [1] (convergence results under a strong convexity assumption on the potential). Then, they prove the convergence of the Schrodinger-Follmer Sampler (SFS) under a regularity assumption (Lipschitzness) and a lower-bound assumption on the ratio between the target and the Gaussian distribution with zero mean and identity covariance matrix. In the case where the lower-bound assumption is not satisfied the authors propose to target another density related to the original distribution and provide non-asymptotic convergence bounds in this case.

[1] Huang, Jiao, Kang, Liao, Liu, Liu -- Schrodinger-Follmer sampler: Sampling without ergodicity

**Requested Changes:**

* Detailed study of the conditions A1 and A2 and the settings under which they hold.

* Experiments to study the tightness of the bounds and justify the modified SFS algorithm.

* A more detailed discussion on the comparison with Langevin-based samplers.

**Strengths And Weaknesses:**

STRENGTH:

* The paper provides novel bounds regarding the SFS. This is an interesting and highly relevant subject.
* The authors clearly discuss the differences between their contribution and the ones of Huang.

WEAKNESS :

* The first and main weakness of this paper is that the conditions under which the convergence occurs are too strong. Indeed, let \mu be a simple Gaussian distribution of the form $\mathcal{N}(0,\alpha)$ (I consider the one dimensional setting). Then $f(x) \propto \exp[-(1/(2\alpha)-1/2) x^2]$. If $\alpha > 1$ then $1/(2\alpha)-1/2<0$ and then $f$ and $\nabla f$ are not Lipschitz continuous (but only locally Lipschitz). If $\alpha <1$ then $1/(2\alpha)-1/2>0$ and then there is no lower bound. Hence, the only case where Theorem 3.1 and Theorem 3.2 correspond to the setting $\alpha = 1$. It seems hard to find relevant settings in which one can apply Theorem 3.1 or Theorem 3.2. From my understanding (A1) and (A2) imply that the target has exactly the same tail as a Gaussian target which is very limiting. Hence, the result of interest of the paper is Theorem 3.3 but this Theorem deals with a variant of SFS which does not correspond to the one used in practice. There is no practical study to show if this alternative SFS scheme is competitive with the original one.

* The claims in Remark 3.4 are misleading. The dissipativity condition outside of a ball allows to deal with Gaussian mixtures contrary to what the authors state "However, these conditions may not hold
for models with multiple modes, for example, Gaussian mixtures, where their potentials are not convex and the log Sobolev inequality may not be satisfied". It is true that the log-Sobolev inequality is not satisfied but the dissipativity and convexity at infinity correspond to weaker conditions. In addition, the condition of convexity outside of a ball leads to convergence rate which are independent of the dimension and only dependent on the non-convexity radius [1,2], i.e. the remark that "implying that the Langevin samplers suffers from the curse of dimensionality" should be tamed. I think a proper comparison with the Langevin samplers is missing in the paper.

* The paper lacks experiments (even in toy cases) to illustrate settings under which their results apply. Such experiments could also point out to the advantages/limitations of the obtained bounds (are they tight? Are they significant numerically?).

OTHER REMARKS:

* The paper is not very well-written. There are many typos (Lagevin instead of Langevin in the introduction, exiting instead of existing in the introduction, egordicity instead of ergodicity in the introduction for instance). The authors could take more time to explain what is the reasoning behind the drift appearing in SFS (this is the gradient of the optimal potential in the Schrodinger bridge problem).

* The claim that "SFS outperforms the existing samplers based on ergodic samplers based on ergodicity" is quite misleading. In [3] the SFS algorithm is evaluated in small dimensional settings and is yet to be compared to HMC or other samplers in more difficult tasks. In particular, the drift term might have a very bad behavior w.r.t the dimension. Indeed, contrary to the Langevin setting where it only involves $U$, the SFS involves $\mathrm{e}^{U}$. The behavior of the exponential of the potential (i.e. the unnormalized density) might behave badly with the dimension (numerical instabilities etc.).

* The authors could explain that (5) is simply an integration by part (or the Stein identity).

* In Theorem 3.1 and the other theorems it could be interesting to consider the impact of setting t=T instead of t=1 for the final time. Indeed, I suspect that setting the time T might have a high practical impact. This can be justified theoretically looking at the conditions A1 and A2. Roughly speaking the time T could be chosen so that A1 and A2 are satisfied in this case (in my previous Gaussian example this amounts to choosing T=\alpha). However, this fix for the Gaussian setting does not work if the two Gaussians are slightly biaised (or if the target density does not admits Gaussian tails exactly).

* The solution proposed Theorem 3.3 is interesting but the authors should have provided a more detailed study of this new algorithm.

* In the appendix could the authors recall what are the processes or at least give a reference to the relevant equations?

[1] Eberle -- Reflection couplings and contraction rates for diffusions
[2] De Bortoli, Durmus -- Convergence of diffusions and their discretizations: from continuous to discrete processes and back
[3] Huang, Jiao, Kang, Liao, Liu, Liu -- Schrodinger-Follmer sampler: Sampling without ergodicity

---

### Review · Reviewer_mkoi · 2022-06-14

**Summary Of Contributions:**

This paper studies the Schrödinger-Föllmer sampler (SFS) whose aim is to sample according to a probability measure $\mu$ without convexity assumption. Unlike schemes based on the Langevin diffusion, the SFS generates samples in finite time.
The proposed methodology is theoretically grounded and the authors derive guarantees to ensure the correct behavior of the approach.
In particular, the paper provides non-asymptotic error bound of SFS in the Wasserstein two distance by making assumptions on the Radon-Nikodym derivative of $\mu$ with respect to the standard Gaussian distribution.

**Broader Impact Concerns:**

This work being of theoretical and methodological nature, therefore I agree with the authors that it is not necessary to include the Broader Impact Statement.

**Requested Changes:**

The strength of the article would be enhanced by a better presentation of the work. In particular, the results lack of quantifiers to facilitate reading and some notations are not introduced such as $\mathcal{P}(\mathbb{R}^d)$.
Further, maybe the authors should make clickable assumptions to help the reader.
In both assumptions used, the regularity on the function $f$ is missing, for example it is not known whereas $f$ is twice differentiable however the hessian is used several times.
In Remark 2.1-(ii), the $\lim_{R\to\infty}\sup_{\|x\|\ge R}$ should be replaced by $\sup_{x\in\mathbb{R}^d}$ for more clarity.
Moreover, I'm not convinced by point (ii), perhaps some further clarification is needed.
Regarding the typographies, I didn't notice many, in the introduction maybe *import* should be replaced by *important*, *Lagevin* by *Langevin* and *duo to* page 13 by *due to*.

**Strengths And Weaknesses:**

The paper provides mathematical results on the interesting topic of sampling random variables using SFS. Moreover, the authors prove the good behavior of this scheme under some Radon-Nikodym derivative conditions without assuming convexity of the potential function.
The paper is relevant, however I think the authors should discuss more about the possible extensions to their work.
In addition, the standard Gaussian case seems restrictive, would it be possible to consider other distributions such as non-centered Gaussian with any covariance?
The paper would gain in clarity if the conditions were more discussed. For example, it would be interesting to further motivate the use of these assumptions through examples of machine learning problems satisfying them.
Although the calculations seem correct, I have the impression that the second moment of $\mu$ should be taken into account in the bounds, because it also depends on the dimension. Thus, it would be better to make the lower bound explicit without the Bachmann-Landau notation.
Finally, I have concerns about the structure of the paper, I am not convinced by the general presentation and formatting of the results. Clarity would be improved by clearer statements about the assumptions used for the different results.

---

### Review · Reviewer_3sdp · 2022-06-15

**Summary Of Contributions:**

The paper studies a sampler based on an Euler-Maruyama approximation of the Schrodinger-Follmer diffusion process (eq 1). This process and the related sampler have the appealing property that they are insensitive to the scale of the target distribution.

The paper shows non asymptotic convergence results in Wasserstein distance (Thm 3.1 and 3.2) between the law of the sampler's output and the target distribution, in specific conditions (boundedness below, and eventually above, of the density ratio between the target and the standard Gaussian; Lipschitzness and smoothness of the this ratio - Assumptions A1 and A2)







**Requested Changes:**

Here are some remarks about the writing and clarity. All of them are easy to address and thus I expect them to be.

Clarifications:
- C2 holds if the drift term is bounded: are you sure ? take b(x, t) = min(1, sqrt(x)) (bounded, independent of t). It does not seem to satisfy C2.
- A1 can you cite the precise results in the papers you indicate?
- In proof of Lemma A1, 2nd step, you could indicate that you use Jensen's inequality (since you detail the arguments for the first and last step)
- one can easily deduce the growth condition and Lipschitz/Hölder continuity > can you cite more precisely the result in Huang et al 2021 ?  Also the paper you cite was not peer reviewed yet, so it's unclear how reliable the result is.
- Where does (13) come from ?
- In the proof of A4, can you recall that s is the stepsize and t_k defined accordingly
- In the equation below (15) you use Young inequality 2ab \leq a^2 + b^2, you could mention it, or directly state that you use (a + b)^2 \leq 2 a^2 + 2b^2 (instead of developing the square in the firs tline and using Young.
- Last equation in the proof of A4, can you detail birefly? How do you not get an exponential factor, since the majoration is of the form u_{k+1} \leq (1 + s) u_k + ... ?
- In A3 how do you go from first inequality to the second, with the s appearing ? it seems you use $ab \leq s a^2 + b^2$, and Jensen's inequality how?

Typos and formulation:
- Please number all of your equations to make future references to the paper (and reviewing) easier.
- that the potential U (x, t) is uniformly strongly convex > add "in x"
- M is one finite and positive constant. > M is a positive constant (if it is infinite the inequality cannot be satisfied)
- Eq 4 repeats the equation at the end of page 1 (please number of your equations for easier communication in the future), what is the difference in this paragraph?
- Eq 5 you seem to use the particular form of f, but it is not made clear in the paragraph above. I suggest to replace "to this end" by "in that case".
- Beginning of P4, this is a repetition of the claim above equation 2. Please reformulate.
without requiring the strongly convexity > without requiring strong convexity
- the uniformly strongly convexity requirement > the uniform strong convexity requirement (many occurrences, please fix all)
- exiting samplers > existing
- More backgrounds on the SFD process see > *For more background* (no 's') ...
- Once V is twice differentiable > If V is twice differentiable ?
- Hess : this reads as H times e times s .... Use \DeclareMathOperator{\Hess}{Hess}  and then call \Hess, or just \nabla^2.
- contant
- since drift term : since the drift term. Also this phrase is missing a conclusion: since the drift term is invariant, what?
- May not to be known: may not be known, or even "is unknown".
- strongly uniform convexity assumption > the uniform strong convexity
- admits the unique strong solution > a unique strong solution
- The Wasserstein of order > the Wasserstein distance of order d. In fact it seems you only use the one of order d, so you can efine directly without using d.
- has the finite upper bound > has a finite upper bound (or, more classical in my opinion, "is upper bounded")
- and set ε= > and ε=
- relies a restricted assumption > relies on a restricted assumption
- I don't think that "it yields that" is correct
- The first sentence in the paragraph above (14) is incorrect.

Notation:
\tilde b_m could be b_m (the same way d_m and e_m are the empirical equivalent of d and e)

References:
- Some references are wrong, ie there are just two authors in "Alain Durmus, Eric Moulines, et al. Nonasymptotic convergence analysis for the unadjusted langevin algorithm."
- Proper capitalization is also missing in names such as {Langevin}, {Sobolev}, Follmer, Monte Carlo etc.
- There is a wrong encoding in Stochastic processesâĂŤmathematics

**Strengths And Weaknesses:**

I have proofread the computations and did not pick up any error. The applicability and empirical performance of the method are not assessed, but it is my understanding that this is not a requirement of TMLR.

In remark 2.1 ii,the authors should give examples of such potentials to justify the interest, since the assumptions A1 and A2 seem quite stringent to me at first read.

---

### Note · Authors · 2022-06-30

I have read and agree with the venue's withdrawal policy on behalf of myself and my co-authors.